# Linking Groundwater to Surface Discharge Ecosystems: Archaeal, Bacterial, and Eukaryotic Community Diversity and Structure in Quebec (Canada)

**DOI:** 10.3390/microorganisms11071674

**Published:** 2023-06-27

**Authors:** Benjamin Groult, Vicky St-Jean, Cassandre Sara Lazar

**Affiliations:** Department of Biological Sciences, University of Québec at Montréal (UQAM), C.P. 8888, Succ. Centre-Ville, Montréal, QC H3C 3P8, Canada; benjamingroult@outlook.fr (B.G.);

**Keywords:** groundwater, surface discharge, archaea, bacteria, eukaryote, microbial community

## Abstract

Aquifer systems are composed of water flowing from surface recharge areas, to the subsurface and back to the surface in discharge regions. Groundwater habitats harbor a large microbial biomass and diversity, potentially contributing to surface aquatic ecosystems. Although this contribution has been widely studied in marine environments, very little is known about the connection between underground and surface microbial communities in freshwater settings. Therefore, in this study, we used amplicon sequencing to analyze the archaeal, bacterial, and eukaryotic community diversity and structure in groundwater and surface water samples, spanning the vast regions of the Laurentides and Lanaudières in the Quebec province (Canada). Our results show significant differences between subsurface and surface taxa; with more fungi, Amoebozoa, and chemolithoautotrophic prokaryotes involved in nitrogen-, sulfur-, and iron-cycling dominating the underground samples; while algae, ciliates, methanogens, and Actinobacteria dominate the surface discharge waters. Microbial source tracking suggested that only a small portion of the microbial communities in the groundwater contributed to the surface discharge communities. However, many taxa were shared between both habitats, with a large range of functional diversity, likely explaining their survival in both subsurface and surface water ecosystems.

## 1. Introduction

Underground aquifer systems are important habitats since they are a major source of drinking water, for humans and agriculture [1]. Aquifers are constituted of water (groundwater) seeping from surface recharge areas, flowing through particles or fractured rock layers in the terrestrial subsurface [2]. This groundwater can subsequently be discharged back in surface aquatic environments, such as lakes, rivers, or streams. As such, aquifers function as open systems, with water flowing once in the subsurface. Therefore, groundwater recharge and discharge mechanisms mediate constant dynamic interactions between surface and subsurface habitats [3]. Aquifers are composed of varying geological (e.g., unconsolidated sands, gravel, permeable consolidated rocks such as sandstone or limestone) and geochemical (e.g., different pH, oxygen, or nutrient concentrations) conditions [4]. These ecosystems harbor a high microbial biomass, as well as a considerable diversity [5], despite the difficult conditions found in underground environments (e.g., dark, oligotrophic, sometimes anoxic).

When groundwater flows back to the surface in discharge regions, its microbial community will contribute to the existing community, although environmental conditions are significantly different in surface aquatic habitats. Because very distinct taxa are found in groundwater ecosystems, they may play an interesting role in surface habitats, when they survive. Most of the studies on microbial groundwater discharge have been carried out in marine systems (submarine groundwater discharge, SGD), where the authors show, for example, that nutrient inputs from SGD are very important [6]. Previous investigations on SGD have highlighted significant contributions of subsurface bacteria to coastal water bodies, with increases in facultative anaerobes [7], as well as bacterial activities likely affecting the chemical composition of SGD discharging into the water column [8]. However, these results also show a significant impact of salinity on bacterial community composition, which cannot be applied to freshwater habitats. There exists very little information and research on microbial community discharge from groundwater to freshwater surface habitats. One study focused on the flow of water from groundwater to hypersaline lake sediments in New Mexico, where the authors showed that salinity and temperature differences between subsurface and sediment niches seemed to have selected very different microbial communities’ diversity [9]. Furthermore, one other looked at water exchanges between groundwater and surface streams, showing that one-third of the microbial communities were similar between habitat types [10].

Thus, in this biogeographical study, we sought to determine the impact of groundwater microbial communities on freshwater aquatic surface discharge environments, in the Laurentides and Lanaudière regions (Quebec province, Canada). These vast areas contain a wide range of different land topographies, watersheds, hydrographic characteristics, plants, agriculture, and soil pedology [11]. They are rich with water, whether underground with aquifer systems characterized by different geological and geochemical conditions, or surface aquatic environments and wetlands [11]. Therefore, they were ideal zones for a groundwater discharge survey in freshwater systems. We sampled groundwater and surface lakes, rivers, and streams spanning 11,500 km^2^, as part of the provincial PACES program (groundwater knowledge acquisition project). We used 16S/18S rRNA gene amplicon sequencing to analyze community composition and structure for the archaeal, bacterial, and eukaryotic communities in both the groundwater and surface samples.

This study is one of the first to analyze microbial discharge of groundwater in freshwater ecosystems on a large scale, in areas with a large variety of subsurface and surface aquatic environments. Furthermore, unlike the few existing previous studies, we targeted all three domains of life, therefore including a complete microbial community in both the underground groundwater and surface freshwater habitats. The Laurentides and Lanaudières areas are home to many urban centers, potentially impacted by groundwater discharge and microbial seepage from underground to the surface. Multivariate analyses showed a significant difference in community composition between both habitats. Furthermore, although most taxa were shared, microbial source tracking suggests little input from groundwater to surface discharge regions.

## 2. Materials and Methods

### 2.1. Study Sites and Sampling

Sampling took place in the summer of 2019, part of the PACES project, in the Lanaudières and Laurentides regions in Quebec, Canada. The groundwater samples that we used in this study to compare with aquatic surface samples are part of a bigger groundwater sample collection that was initially published in Groult et al. [12]. Lakes, rivers, and stream water samples were collected the same day as the surrounding groundwater samples and were chosen as likely discharge zones of groundwater flowing back to the surface (S. Gagné personal communication, Figure 1). Six surface samples did not seem connected to any groundwater systems but were still used as representatives of surface aquatic samples (Appendix A). Lake, river, and stream water samples were collected in sterilized polypropylene bottles (Nalgene, Rochester, NY, USA), transported on ice, and stored at 4 °C until filtration in the lab, which was done the same day as sampling. Filtration was carried out using a 0.2 µm polyethersulfone filter (Sartorius, Göttingen, Germany) with 1 L of water. Filters were subsequently stored at −20 °C.

### 2.2. Geochemical Analyses

Water samples were collected in gas-free glass bottles after filtration on a 0.45 µm polyethersulfone filter (Sarstedt, Newton, MA, USA), to measure dissolved organic and inorganic carbon (DOC and DIC). Samples were analyzed with an OI Analytical Aurora 1030W TOC Analyzer (College Station, TX, USA) by using a persulfate oxidation method, at the GRIL (Interuniversity Research Group in Limnology)-UQAM laboratory.

### 2.3. DNA Extraction and Illumina Sequencing

DNA was extracted from the filters using the DNeasy power water kit (Qiagen, Hilden, Germany) according to the manufacturer’s instructions. All extracted DNA samples were stored at −20 °C until further use.

Sequencing was carried out at the CERMO-FC genomic platform (center for excellence in research on orphan disease—foundation Courtois) at UQAM. Archaeal, bacterial 16S, and eukaryote 18S rRNA genes were amplified using the polymerase UCP hiFidelity PCR kit (Qiagen, Hilden, Germany). The V3-V4-V5 region of the archaeal 16S rRNA gene was targeted using the A340F (5′-CCCTACGGGCYCCASCAG-3′) and 915R (5′-GTGCTCCCCCGCCAATTCCT-3′) primer pair [13,14]. The V3-V4 region of the bacterial 16S rRNA gene was targeted using the B341F (5′-CCTACGGGAGGCAGCAG-3′) and B785R (5′GACTACCGGGGTATCTAATCC-3′) primer pair [15,16]. The V5 region of the eukaryote 18S rRNA gene was targeted using the E960F (5′-GGCTTAATTTGACTCAACRCG-3′) and NSR1438R (5′-GGGCATCACAGACCTGTTAT-3′) primer pair [17,18].

PCR amplification was performed under the following conditions: denaturation at 98 °C for 30 s, annealing for 30 s (58 °C for archaea, 57 °C for bacteria, and 61 °C for eukaryotes), extension at 72 °C for 1 min, and final extension at 72 °C for 10 min. We used 40 cycles for the archaea, 35 for the bacteria, and 35 for the eukaryotes. Sequencing was performed using an Illumina MiSeq 2300 and the MiSeq reagent kit v.3 (600 cycles, Illumina). Negative controls for the PCR amplifications of both the bacteria and archaea were sequenced as well. All surface water sequences were deposited on the National Center for Biotechnology Information platform (NCBI) under the BioProject ID PRJNA867732. Groundwater sequences can be found at PRJNA701529.

### 2.4. Sequence Analyses

The obtained sequences were analyzed using the mothur software v.1.44.3 [19] and were classified with the SILVA database v.138.1 [20]. The classification of the archaeal 16S rRNA genes was further conducted with reference sequences from the Bathyarchaeaota [21] and the Woesearchaeota [22], as well as a personal database. Amplicon sequence variants (ASVs) were obtained using mothur v.1.47.0. Rarefaction was carried out using the median sequencing depth method [23], and we only kept samples with more than 1000 sequences. We subtracted the ASVs that were sequenced in the PCR negative control from all samples. In order to focus on the microeukaryotes, we also deleted all ASVs affiliated with the following phyla for the eukaryotic dataset: Animalia, Annelida, Arthropoda, Cnidaria, Mollusca, Nematozoa, Phragmoplastophyta, Platyhelminthes, Tardigrada, and Vertebrata.

### 2.5. Statistical Analyses

Shannon diversity indices (α-diversity), chao richness indices, and Shannon evenness indices were calculated using mothur. All indices were compared between surface and subsurface groundwater samples using a Wilcoxon test with the wilcox.test function in R [24] v.4.1.2. Community composition (β-diversity) was analyzed with principal coordinate analysis (PCoA) run in mothur, with rarefied ASV tables and a Bray–Curtis dissimilarity distance matrix. Homogeneity of molecular variance (homova) was used on the distance matrix, to test whether there were significant differences within the surface and subsurface water samples. To test whether community composition varied significantly between surface and subsurface water samples, we used permutational multivariate analyses (PERMANOVA) on the rarefied ASV tables in R, with the adonis function of the vegan package. Significantly different taxa (genus level) between sample groups were identified using linear discriminant effect size (LEfSe) analyses with the online tool from the Huttenhower lab (https://huttenhower.sph.harvard.edu/lefse/, accessed 21 June 2023). Unique and shared genera between surface and groundwater samples were determined in mothur. We used distance-based redundancy analysis (db-RDA) to correlate DOC or DIC with community composition. ASV tables were transformed and used to calculate a Bray–Curtis dissimilarity distance matrix. Environmental variables were transformed using log(x + 1). The db-RDA was applied to the distance matrix and the explanatory variables using the capscale function of the vegan package in R, and significance of explanatory variables was assessed with the anova function in R with 200 permutations. The contribution of each significant variable to community composition was determined using variance partitioning with the varpart function of the vegan package in R. We compared DOC and DIC concentrations between surface and groundwater samples using a Wilcoxon test. Finally, we used the FEAST method (fast expectation-maximization for microbial source tracking) [25] to estimate the source of microbial communities from the groundwater to surface aquatic waters, using raw ASV tables without transformation.

## 3. Results

### 3.1. Microbial Gene Diversity

The surface samples were dominated by Woesearchaeota and Halobacteria at the phylum level for the archaea (Figure 2a), with methanogenic genera (*Methanobacterium*, *Methanosaeta*, *Methanoregula,* or *Methanosarcina*), and Woesearchaeota subgroups 5a, 5b and 8 the most abundant at the genus level (Figure 2b). The surface bacteria were composed mostly of Actinobacteria and Proteobacteria (Figure 3a), and unclassified (unc.) Sporichthyaceae, *Rhodoferax*, *Limnohabitans*, *Rhodoluna*, *Polynucleobacter*, and *Mycobacterium* at the genus level (Figure 3b). The surface eukaryotes were composed mostly of Ochrophyta, Chryptophyceae, and Diatomae (Figure 4a), and of unc. Chrysophyceae, *Cryptomonas*, *Synura,* and *Uroglena* at the genus level (Figure 4b).

The subsurface groundwater samples were dominated mainly by Woesearchaeota and Nanoarchaeota at the phylum level for the archaea (Figure 2a), with Woesearchaeota subgroup 24, and candidatus (cand.) Nitrosotalea, cand. Methanoperendens, and GW2011_GWC1_47_15 Nanoarchaeota the most abundant at the genus level (Figure 2b). The groundwater bacteria were composed mostly of Proteobacteria (Figure 3a), and *Gaillonella*, unc. Oxalobacteraceae, CL500-29_marine_group, Acinetobacter, *Nitrospira* and many other taxa at the genus level (Figure 3b). The groundwater eukaryotes were composed mostly of Ascomycota, Basidiomycota, Ocrophyta, Ciliophora, and Cercozoa (Figure 4a), and unc. Sordariomycetes, unc. Ascomycota, *Scuticociliatia*, unc. Capnodiales, unc. Lichtheimiaceae, unc. Chrysophyceae, and unc. Cercozoa at the genus level (Figure 4b).

### 3.2. Microbial α-Diversity Indices and Comparison

For the surface samples, archaeal α-diversity indices ranged from 3.93 to 8.15, 3.96–8.46 for the bacteria, and 2.99–5.96 for the eukaryotes. Richness ranged from 253.045 to 1,547,989 for the archaea, 1296.52–66,317.13 for the bacteria, and 243–45,191 for the eukaryotes. Evenness ranged from 0.523 to 0.898 for the archaea, 0.59–0.93 for the bacteria, and 0.41–0.75 for the eukaryotes (Appendix A). For the subsurface groundwater samples, archaeal α-diversity indices ranged from 1.02 to 7.31, 1.83–7.84 for the bacteria, and 0.9–6.05 for the eukaryotes. Richness ranged from 180.89–9,882,141 for the archaea, 82–3,343,699 for the bacteria, and 16–128,207 for the eukaryotes. Evenness ranged from 0.211 to 0.852 for the archaea, 0.244–0.907 for the bacteria, and 0.17–0.87 for the eukaryotes.

We compared indices between surface and subsurface samples. For the archaea, the diversity indices were not significantly different (*p* = 0.842), the richness indices were significantly different (*p* = 0.002278) with the groundwater indices being higher than the surface indices, and the evenness indices were significantly different (*p* = 0.001099) with the surface indices being higher than the groundwater indices. For the bacteria, the diversity indices were not significantly different (*p* = 0.1966), the richness indices were significantly different (*p* = 0.03631) with the groundwater indices being higher than the surface indices, and the evenness indices were not significantly different (*p* = 0.967). For the eukaryotes, the diversity indices were significantly different (*p* = 6.627 × 10^−6^) with the surface indices being higher than the groundwater indices, the richness indices were significantly different (*p* = 1.758 × 10^−6^) with the surface indices being higher than the groundwater indices, and the evenness indices were not significantly different (*p* = 0.5776).

### 3.3. Microbial Community Composition

The PCoA plots for all 3 domains showed that most of the surface samples clustered together and showed the same for the subsurface groundwater samples (Figure 5a–c). This was supported by the PERMANOVA tests showing that habitat (surface or subsurface) was a significant environmental factor explaining 4.22% of the variance in the archaeal community, 6.32% in the bacterial community, and 6.51% in the eukaryotic community (Table 1). The homova tests highlighted significant differences in the variation within the surface and subsurface sample clusters, with the subsurface groundwater samples having a higher variation than the surface samples (Appendix A).

For the archaea, this significant difference between habitats was explained using LEfSe at the phylum level by Halobacteria, Euryarchaeota, and Bathyarchaeota that were significantly higher in the surface samples, and Crenarchaeota and Thermoplasmatota that were higher in the subsurface samples (Appendix A). At the genus level, methanogenic genera (*Methanosaeta*, *Methanosarcina*, *Methanobacterium*, *Methanoregula*), Woesearchaeota subgroups 8 and 10 and Bathyarchaeota subgroup 6 were significantly higher in the surface samples, whereas cand. Nitrosotalea, cand. Methanoperedens, *Nitrosoarchaeum*, and unc. Thermoplasmatota were higher in the subsurface samples (Figure 6a). For the bacteria, Actinobacteriota were significantly higher in the surface samples, whereas Proteobacteria, Nitrospirota, and Verrucomicrobiota were higher in the subsurface samples (Appendix A). At the genus level, the hgcI-clade, unc. Sporichtyaceae, *Polynucleobacter*, *Limnohabitans*, and *Mycobacterium* were significantly higher in the surface samples, whereas unc. Gallionellaceae, *Gaillonella*, *Staphylococcus*, *Nitrospira*, and *Sphingomonas* were higher in the subsurface samples (Figure 6b). For the eukaryotes, Ochrophyta, Cryptophyceae, Chytridiomycota, Diatomea, Chlorophyta, Dinoflagellata, and Rotifera were significantly higher in the surface samples, whereas Ascomycota, Cercozoa, Basidiomycota, and Amoebozoa were higher in the subsurface samples (Appendix A). At the genus level, unc. Chrysiphyceae, *Cryptomonas*, *Ploimida*, *Pythium*, *Uroglena*, unc. Choreotrichia, and unc. Chtyridiomycetes were significantly higher in the surface samples, whereas unc. Ascomycota, unc. Cercozoa, *Neosphaeosphaeria*, *Spumella*, *Vermamoeaba*, *Rhodotorula*, and *Tetrahymena* were higher in the subsurface samples (Figure 6c).

### 3.4. Shared and Unique Taxa in the Surface and Subsurface Samples

When combining all surface and subsurface samples at the phylum level, for the archaea, 86.7% of the taxa were shared between both habitats (Appendix A), with no unique taxa in the surface samples, and 13.3% unique taxa in the subsurface samples. When looking at each site separately, most taxa were also shared between both habitats, and only four sites showed unique taxa in the subsurface samples. Woesearchaeota and Nanoarchaeota were the major shared taxa, and Hydrothermarchaeota was the dominant taxon only found in the subsurface samples (Appendix A). At the genus level, when combining all surface and all subsurface samples, 75.2% of the taxa were shared, with 13.1% taxa unique to the surface habitat and 11.7% unique to the subsurface habitat (Appendix A). When looking at each site separately, the percentage of shared taxa between habitats was lower. Unc. Woesearchaeales, Woesearchaeota subgroups 24, 5a, 5b, and 8, and CG1-02-32-21 Micrarchaeota were the major shared taxa (Appendix A). Unc. Methanocellaceae, Rice Cluster I and II, *Methanocella*, unc. Methanoregulaceae, and Woesearchaeota subgroup 9 were the dominant taxa only found in the surface samples. Unc. Methanoperedenaceae, cand. Methanoperedens, unc. Methanomassoliiococcales, unc. Nitrosotaleaceae, unc. Nitrospumilaceae, unc. Hydrothermarchaeles, and Marine Benthic Group A (MBG-A) were the dominant taxa only found in the subsurface samples.

When combining all surface and subsurface samples at the phylum level, for the bacteria, 76.6% of the taxa were shared between both habitats (Appendix A), with 6.3% unique taxa in the surface samples, and 15.6% unique taxa in the subsurface samples. When looking at each site separately, most taxa were also shared between both habitats, and most sites had more unique taxa in the surface than in the subsurface. Actinobacteriota and Proteobacteria were the major shared taxa; Deferribacterota, Armatimonadota, Campilobacterota, Edwardsbacteria, Fusobacteriota, and Synergistota were the dominant taxa only found in the surface samples, and Methylomirabilota, DTB120, Entotheonellaeota, GAL15, Latescibacterota, NKB15, and RCP2-54 were the dominant taxa only found in the subsurface samples (Appendix A). At the genus level, when combining all surface and subsurface samples, 61.1% of the taxa were shared, with 17.9% taxa unique to the surface habitat and 21% unique to the subsurface habitat (Appendix A). Unc. Comamonadaceae, *Rhodoferax*, *Mycobacterium*, unc. Sporichtyaceae, unc. Xanthobacteraceae, unc. Burkholderiales, CL500-29_marine_group, Flavobacterium, and *Nitrospira* were the major shared taxa (Appendix A). *Limnobacter*, cand. Planktoluna, GKS98_freswhater_group, 966-1 Nitrosomonadaceae, *Chitinibacter*, and *Longivirga* were the dominant taxa only found in the surface samples. Pla4_lineage, unc. Rokubacteriales, SAR202_clade, OLB14 Chloroflexi, unc. Acidiferrobacteraceae, and unc. Verrucomicrobiota were the dominant taxa only found in the subsurface samples, as well as *Nitrobacter*, *Gallionella*, and *Sulfuricella*.

When combining all surface and subsurface samples at the phylum level, for the eukaryotes, 87.18% of the taxa were shared between both habitats (Appendix A), with 7.69% unique taxa in the surface samples, and 2.56% unique taxa in the subsurface samples. When looking at each site separately, more than half of the taxa were also shared between both habitats. Ochrophyta, Ascomycota, Basidiomycota, Cercozoa, and Ciliophora were the major shared taxa; Diatomea, Dinoflagellate, and Aphelida were the dominant taxa only found in the surface samples; and Amoebozoa and Mucoromycota were the dominant taxa only found in the subsurface samples (Appendix A). At the genus level, when combining all surface and subsurface samples, 59.7% of the taxa were shared, with 28.89% taxa unique to the surface habitat and 11.5% unique to the subsurface habitat (Appendix A). When analyzing the different sites separately, the fraction of shared taxa ranged from 23.5–60.93% and the fraction of unique taxa in the surface samples was highest (5.8–65.9%). Unc. Chrysophyceae, unc. Ascomycota, *Cryptomonas*, unc. Cercozoa, *Neophaeosphaeria*, and unc. Oligomenophorea were the major shared taxa (Appendix A). *Choreotrichia*, *Ploimida*, and *Halteria* were the dominant taxa only found in the surface samples. *Vermamoeaba*, *Bannoa*, *Gymnophrys*, *Peritrichia*, and unc. Pleosporales were the dominant taxa only found in the subsurface samples.

### 3.5. Microbial Community Correlation with Environmental Parameters

DIC and DOC concentrations were significantly different between surface and subsurface samples (*p* = 6.156 × 10^−6^ and 4.593 × 10^−5^), with the subsurface DIC higher than the surface DIC, and the surface DOC higher than the subsurface DOC (Appendix A). For all three domains, based on the db-RDNA analysis, both DIC and DOC were significant environmental variables correlated with archaeal, bacterial, and eukaryote community composition (Table 2). For all three domains, DOC was mostly correlated to the surface samples, while DIC was correlated with the subsurface samples (Figure 7a–c). Variance partitioning indicated that DIC explained 1.2% of the archaeal variance, 1.9% of the bacterial variance, and 2.1% of the eukaryotic variance. DOC explained 0.4% of the archaeal variance, 0.8% of the bacterial variance, and 0.9% of the eukaryotic variance (Appendix A).

### 3.6. Contribution of Groundwater Microbial Communities to the Surface Communities

Microbial source tracking showed that when using aquifers likely discharging water into nearby surface aquatic environments, up to 3.25% of the archaeal communities in the groundwater contributed to the surface communities, 12.48% of the bacterial communities, and 9.39% of the eukaryotic communities (Figure 8a–c).

## 4. Discussion

### 4.1. Microbial Community Differences between Subsurface and Surface Aquatic Habitats

Our study showed that for the eukaryotic communities, the number of species was significantly higher in surface waters compared to groundwater samples, and community composition also differed. We could not find previous studies directly comparing microeukaryotic community diversity between groundwater and freshwater discharge systems. However, there are many studies on microeukaryote communities in surface freshwater habitats, in lakes and rivers. Mostly phytoplankton, ciliates, dinoflagellates, and algae are found in surface water surveys [26,27,28], while fungi, Metazoa, Ciliophora, Cercozoa, and Amoebozoa are detected in groundwater studies [12,29,30]. The higher species richness in surface habitats is likely linked to the presence of sunlight and photosynthetic microorganisms which are absent in the dark subsurface ecosystems driven by chemosynthesis, as well as higher organic matter availability in surface systems given the direct link to terrestrial matter [31,32]. This is supported by the higher DOC concentrations in our surface samples and the significant correlation between eukaryotic community structure (β-diversity) and DOC concentrations.

For the prokaryotes (archaea and bacteria), there were no significant differences in taxonomic diversity. However, for both domains, species richness was significantly higher in the groundwater compared to the surface freshwater samples. Most of the prokaryotic community studies on groundwater discharge are carried out in coastal environments [7,8], where water geochemical conditions (such as salinity) and hydrology dynamics are extremely different, rendering comparison with freshwater ecosystems unreliable. The decrease in eukaryotic species richness—many of which are prokaryote predators—is a potential driver of higher prokaryotic species richness in subsurface aquatic habitats [33,34]. Furthermore, the absence of sunlight is likely another driver of higher prokaryotic species richness, as the subsurface microbial community are based on chemosynthesis [35]. A study of photosynthetic and chemosynthetic bacteria in a desert aridity gradient showed that the chemosynthetic bacteria were able to better survive desiccation and starvation, and that their ability to oxidize trace gas conferred an advantage by providing alternate electron donors [36]. Therefore, the higher DIC concentrations and the significant correlation between prokaryotic communities and DIC, the dark oligotrophic conditions found in aquifers leading to a need to find other sources of electron donors and carbon than those found in surface habitats, could result in the observed higher number of prokaryotic species.

For all domains, variation in the composition (β-diversity) of the surface samples was reduced compared to the groundwater samples, suggesting overall a more stable community in the surface waters during the summer season. It is likely that the surface freshwater habitats faced less environmental perturbation than the groundwater samples. In the case of this study, this result probably highlights the heterogeneity of abiotic conditions that microbial communities face in the underground ecosystems (e.g., geological, hydrological, mineralogic, geochemical, or physico-chemical features) [37], whereas conditions in lakes and rivers during the summer fluctuate less [38].

### 4.2. Distinguishing Microbial Taxa in Subsurface and Surface Aquatic Environments

Since habitat type (subsurface or surface) was an environmental variable with a significant effect on community composition for all microbial domains, we delved into the different taxa explaining these differences. As previously shown in aquifers from different regions [12,29,30], groundwater phyla in our study contained significantly more fungi, Cercozoa, and Amoebozoa for the eukaryotes. In general, aquifer fungi are heavily involved in complex organic matter decomposition (such as plant-derived compounds), releasing smaller carbohydrates and nutrients which can be picked up and assimilated by other microbes [39]. The *Gymnophrys* cercozoan, *Vermamoeaba* amoeba, and *Peritrichia* ciliate were taxa unique to the groundwater, all eukaryotes being involved in bacterial predation [30,40,41,42]. Thus, the microeukaryotes seem to be involved in groundwater microbial community interactions, whether providing usable organic carbon molecules, or acting as prokaryote predators.

Furthermore, the subsurface samples contained significantly more nitrogen-cycling archaea and bacteria (cand. Nitrosotalea, cand. Methanoperedens, *Nitrosoarchaeum*, *Nitrospira*), and iron-cycling bacteria (unc. Gallionellaceae, *Gaillonella*) compared to the surface samples, as well as unc. Thermoplasmatota for the archaea, and *Staphylococcus* and *Sphingomonas* for the bacteria. In addition, unique groundwater taxa included unc. Methanoperedenaceae, cand. Methanoperedens, unc. Nitrosotaleaceae, unc. Nitrospumilaceae, unc. Methanomassoliiococcales, and Hydrothermarchaeota for the archaea, and *Nitrobacter*, *Gallionella*, *Sulfuricella*, and unc. Acidiferrobacteraceae for the bacteria. Thus, one methanogenic taxon was associated with the underground habitats (Methanomassiliococcales [43]), and methane production could be linked to the presence of the methane oxidizer Methanoperendens [44]. A large majority of these groundwater-associated or unique taxa have chemolithoautotrophic metabolisms and are involved in ammonia- (*Nitrosotalea* and *Nitrosoarcheum* [45,46]), nitrite- (*Nitrospira* and *Nitrobacter* [47,48]), sulfur- (Acidiferrobacteraceae [49], *Sulfuricella* [50]), iron- (*Gaillonella* [51], Acidiferrobacteraceae), and H_2_-oxidation (Hydrothermarchaeota [52]). As groundwater ecosystems are devoid of light, they rely solely on chemoautotrophy for primary production. Supported by previous microbial community surveys [30,53], and the higher DIC concentrations in our groundwater sites, we show the importance of chemoautotrophic taxa in aquifer environments spanning a vast area in the Quebec region, involved in nitrogen-, sulfur-, and iron-cycling.

Also, many of these taxa are potentially adapted to the specific or difficult settings found in underground ecosystems, such as pH regulation (*Nitrosotalea* and *Nitrosoarcheum* [46], *Gaillonella* [54]), osmotic regulation (Methanomassoliiococcales [43], Thermoplasmatota [55]), oligotrophic conditions (Thermoplasmatota [56], *Sphigomonas* [57]), and arsenic- or heavy-metal-oxidation (Methanomassoliiococcales [43], Hydrothermarchaeota [52], *Gaillonella* [54], *Staphylococcus* [58]). This likely highlights the taxonomic and functional flexibility of microbial communities in fluctuating groundwater environments.

In contrast, the surface water samples contained significantly more Ochrophyta, Cryptophyceae, Chytridiomycota, Diatomea, Chlorophyta, and Dinoflagellata for the eukaryotes, with Diatomea and Dinoflagellate being unique phyla in the surface samples. Most of these eukaryotes were either photoautotrophic algae (*Cryptomonas* [59]), or mixotrophic ciliates involved in algae-, bacteria- and even virus-consumption (*Choreotrichia* [60], *Halteria* [61], *Ploimida* [62]). Surface water samples contained significantly more methanogenic archaea (*Methanosaeta*, *Methanosarcina*, *Methanobacterium*, *Methanoregula*), which is a common observation in lake and river waters [63,64,65], and mostly methanogenic genera (Rice Cluster I and II, *Methanocella*) were unique. In addition, Bathyarchaeota subgroup 6, who potentially possess a microoxic and light-sensing lifestyle [66], was also higher in the surface samples. All prokaryotic taxa that were significantly higher in the surface waters were chemoorganotrophs involved in carbon-cycling (Bathyarchaeota [66], hgcl clade [67], *Polynucleobacter* [68,69], *Limnohabitans* [70,71]). Most bacterial taxa unique to the surface were also chemoorganotrophs (*Limnobacter* [72], *Planktoluna* [73], GKS98_freswhater_group [74], *Longivirga* [75]), with *Chitinibacter* being a chitin-degrading organism [76] which is the major component of fungal cell walls, insect exoskeletons, and crustacean shells. Here, the higher DOC concentrations and the continuous availability of terrestrial organic matter likely selected a heterotrophic-based community.

### 4.3. Links between Subsurface and Surface Aquatic Microbial Communities

Using microbial source tracking, we showed that for all 3 domains, less than 13% of the surface aquatic communities originated from the groundwater discharging into these aquatic systems, with subsurface bacteria contributing the most to surface communities, and archaea the less. Nonetheless, at the phylum level, more than 76% of all taxa for all 3 domains were shared between both habitats, and more than 59% at the genus level, suggesting that many microbial genera dwelling in groundwater habitats can adapt when reaching surface waters. The higher diversity within the bacteria could allow survival of more taxa from one habitat to the other compared to the other domains.

Proteobacteria and Actinobacteriota were the major bacterial shared phyla, and within these phyla, unc. Comamonadaceae, unc. Xanthobacteraceae, and unc. Sporichtyaceae were the major shared families, including a large variety of phenotypic diversity (aerobic chemoheterotrophs, facultative chemolithoautotrophs, anaerobic denitrifiers, alkene- and halogenated-compounds users, Fe(III)-reducers, H_2_-oxidizers, N_2_-fixation, photoautotrophs, photoheterotrophs, and fermenters) and habitats (water and soil) [77,78]. For the archaea, Woesearchaeota and Nanoarchaeota (CG1-02-32-21 Micrarchaeota) were the major shared taxa. Although the Woesearchaeota are widespread, they seem to prefer inland anoxic habitats, and are mostly heterotrophs with metabolic deficiencies and probably live in syntrophy or symbiotically [22]. The Micrarchaeota are small archaea known to physically interact with other archaea and to scavenge amino acids and nucleotides since they seem to lack these biosynthetic pathways. They also prefer low-oxygen environments and are probably involved in carbon- and iron-cycling [79]. Finally, for the eukaryotes, algae and fungi were the most shared taxa between underground and surface environments. The fungi (*Neophaesophaeria*) are widespread in freshwater habitats and are saprotrophs and heterotrophs [80]. The Chrysophyceae algae are oligotrophic lake plankton and can shift from photosynthesis to a heterotrophic lifestyle by food particles and smaller organisms’ ingestion [81]. Therefore, not only the higher taxonomic diversity, but the higher potential functional diversity in the bacteria found in both types of habitats, could explain why microorganisms belonging to this domain seem to survive better when flowing from groundwater to surface discharge waters.

## 5. Conclusions

In this study we carried out a biogeographic survey of microbial communities belonging to the three domains of life in surface groundwater discharge habitats, in comparison to a biogeographic survey of microbial communities in groundwater ecosystems. We showed that diversity, richness, and structure differed between communities from underground and surface environments, for eukaryotes, bacteria, and archaea. The taxa explaining these differences were affiliated mostly with chemoautotrophic species in the groundwater habitats, potentially involved in carbon-, nitrogen-, sulfur- and iron-cycling, and containing stress response pathways likely allowing them to survive in the dark oligotrophic conditions found in the terrestrial subsurface. The surface taxa were mostly affiliated with methanogens and heterotrophic microorganisms, probably correlated with an easier access to fresh organic matter. Finally, our analyses suggested that a small portion of the groundwater taxa contributed to the surface discharge, and bacterial species with a vast potential range of functional diversity seemed to be most shared between habitat types. Future metagenomic-based studies would allow confirmation of this functional diversity and flexibility between groundwater and surface discharge water ecosystems.

## Figures and Tables

**Figure 1 microorganisms-11-01674-f001:**
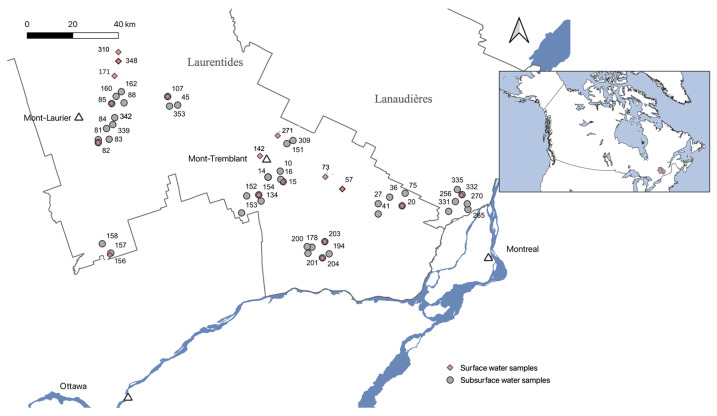
Map of the sampled wells and surface water in the Laurentides and Lanaudières regions in Quebec, Canada (using the Qgis software v.3.28.8). White triangles locate cities in the Laurentides area.

**Figure 2 microorganisms-11-01674-f002:**
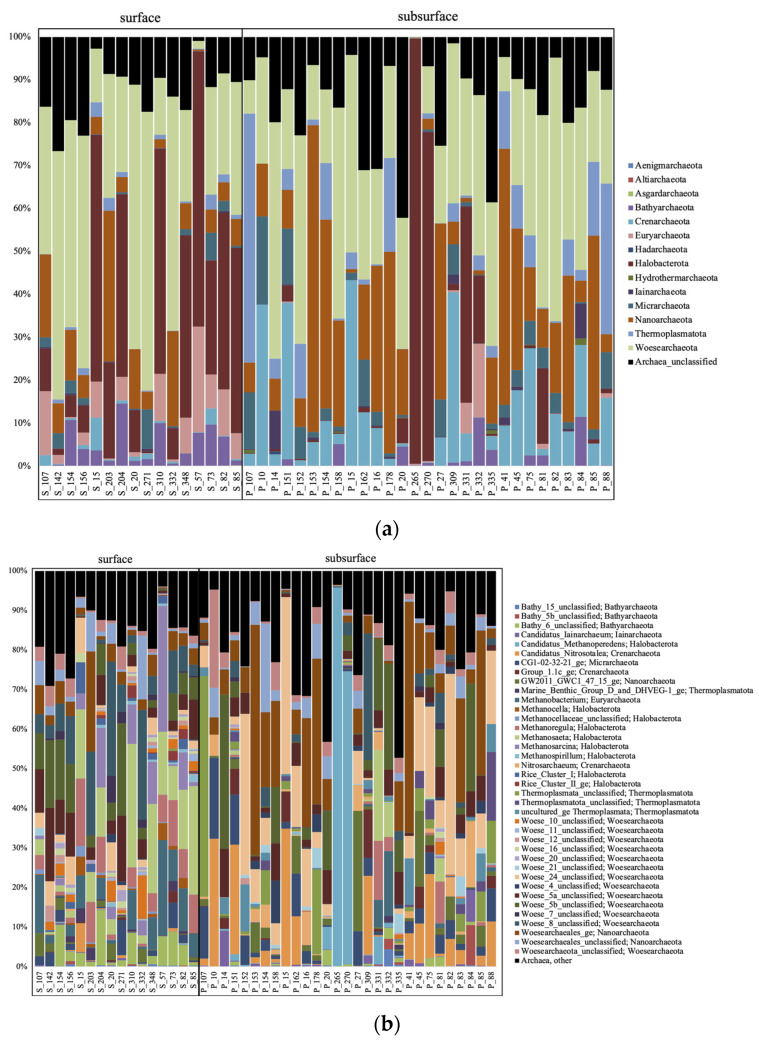
Archaeal taxa based on the 16S rRNA gene diversity at the phylum (**a**) and genus (**b**) levels, represented in percentage of the total number of reads.

**Figure 3 microorganisms-11-01674-f003:**
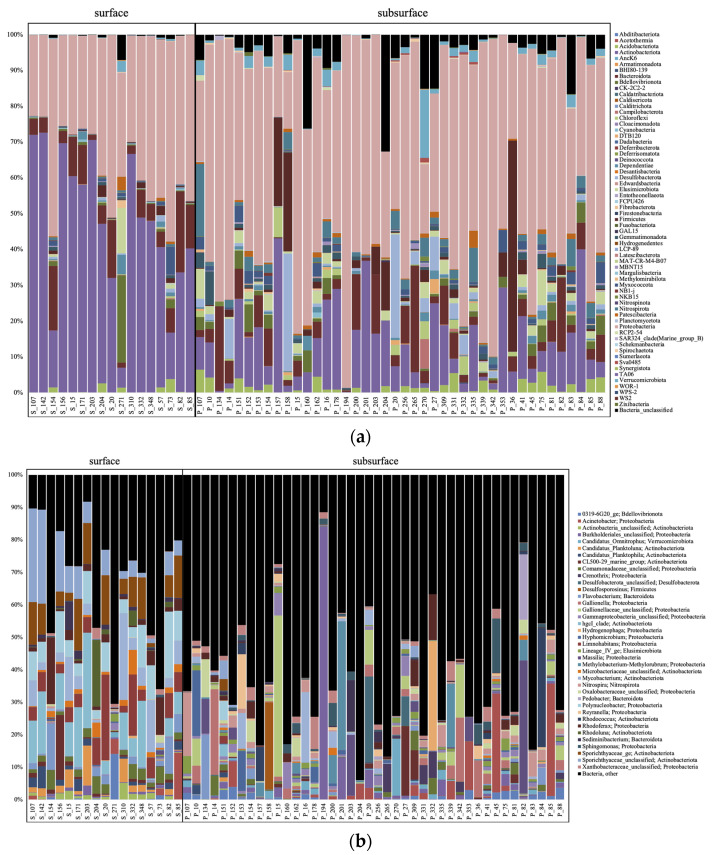
Bacterial taxa based on the 16S rRNA gene diversity at the phylum (**a**) and genus (**b**) levels, represented in percentage of the total number of reads.

**Figure 4 microorganisms-11-01674-f004:**
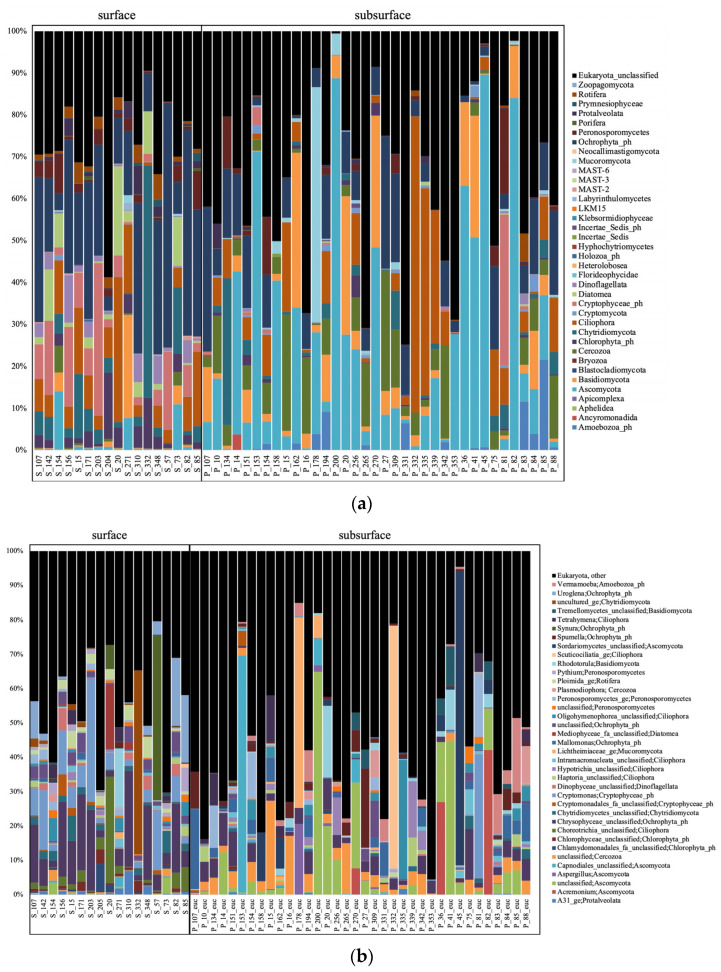
Eukaryotic taxa based on the 16S rRNA gene diversity at the phylum (**a**) and genus (**b**) levels, represented in percentage of the total number of reads.

**Figure 5 microorganisms-11-01674-f005:**
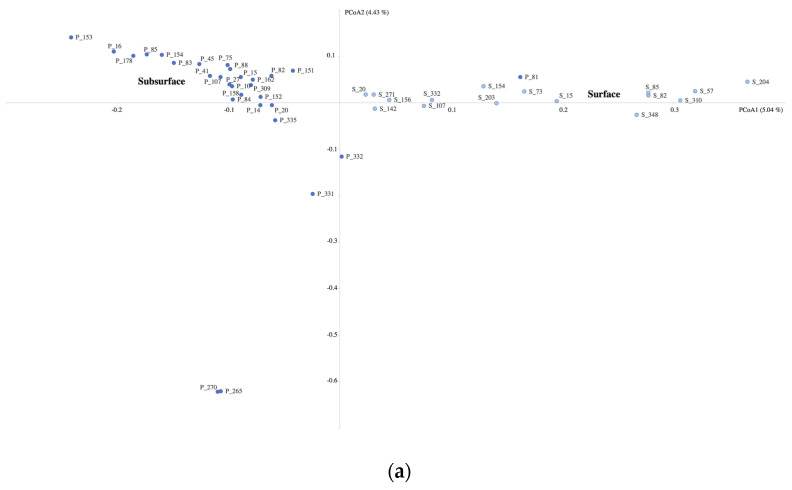
Principal coordinate analysis (PCoA) of the archaeal (**a**), bacterial (**b**), and eukaryotic (**c**) communities, based on a Bray–Curtis dissimilarity matrix. P, groundwater samples; S, surface water samples. Light blue indicates surface samples, and dark blue the subsurface samples.

**Figure 6 microorganisms-11-01674-f006:**
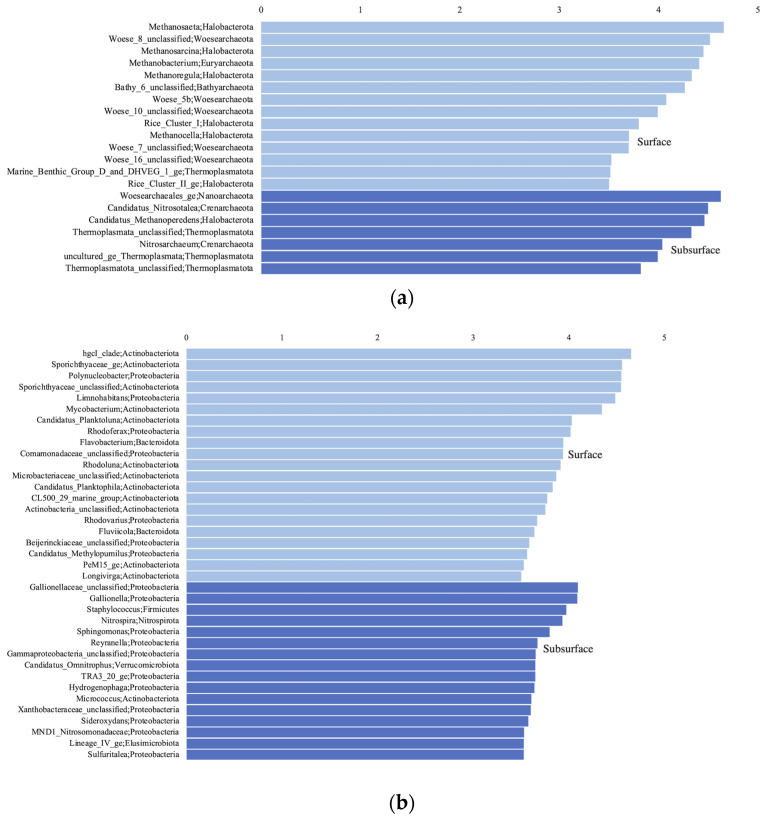
Linear discriminant analysis (LDA) score comparing significantly different archaeal (**a**), bacterial (**b**), and eukaryotic (**c**) genera between the groundwater (subsurface) and surface samples, calculated using the LefSe analysis. Genera with a LDA score > 3.5 are displayed.

**Figure 7 microorganisms-11-01674-f007:**
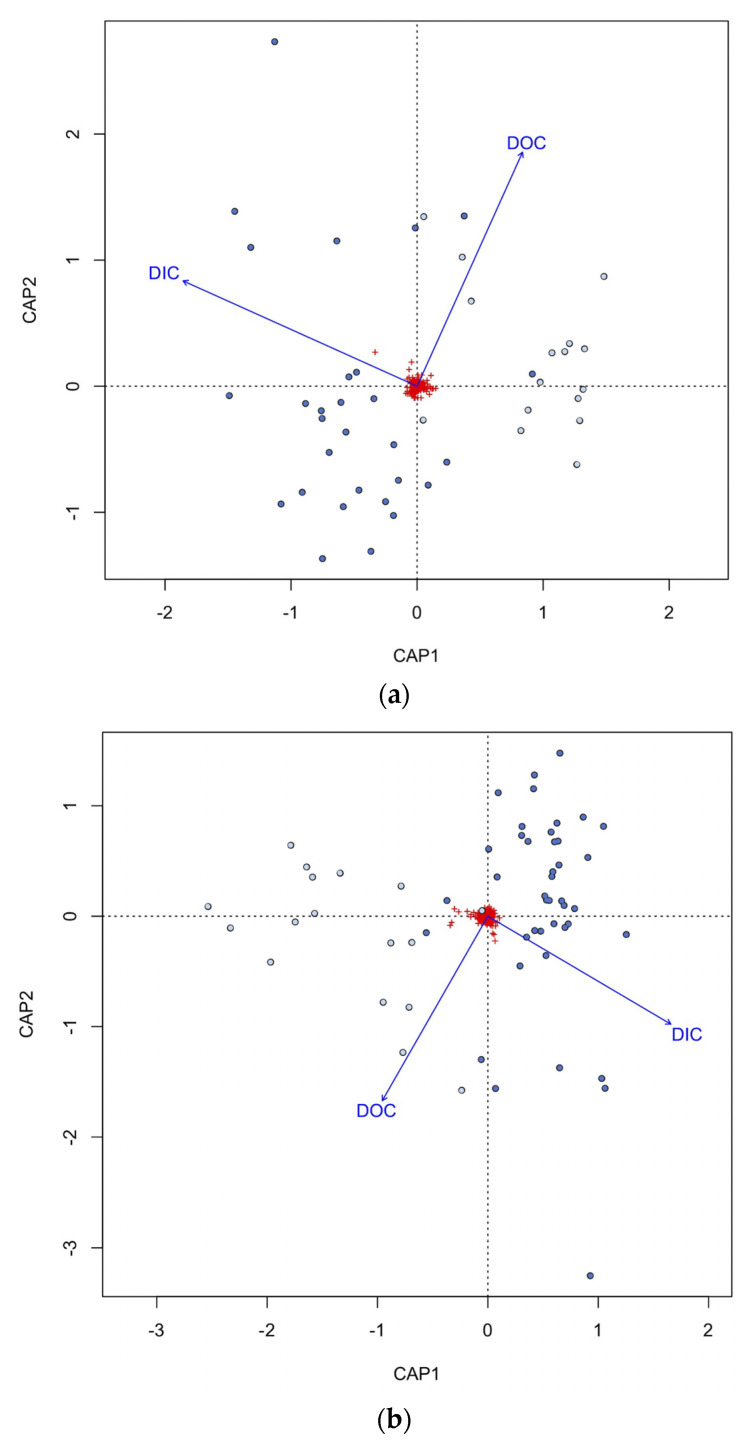
Correlation between the archaeal (**a**), bacterial (**b**), and eukaryotic (**c**) community composition and explanatory factors in the surface and subsurface water samples using db-RDA. Groundwater samples are colored dark blue, and surface water samples are light blue. Red crosses are the ASVs. DOC, dissolved organic carbon; DIC, dissolved inorganic carbon.

**Figure 8 microorganisms-11-01674-f008:**
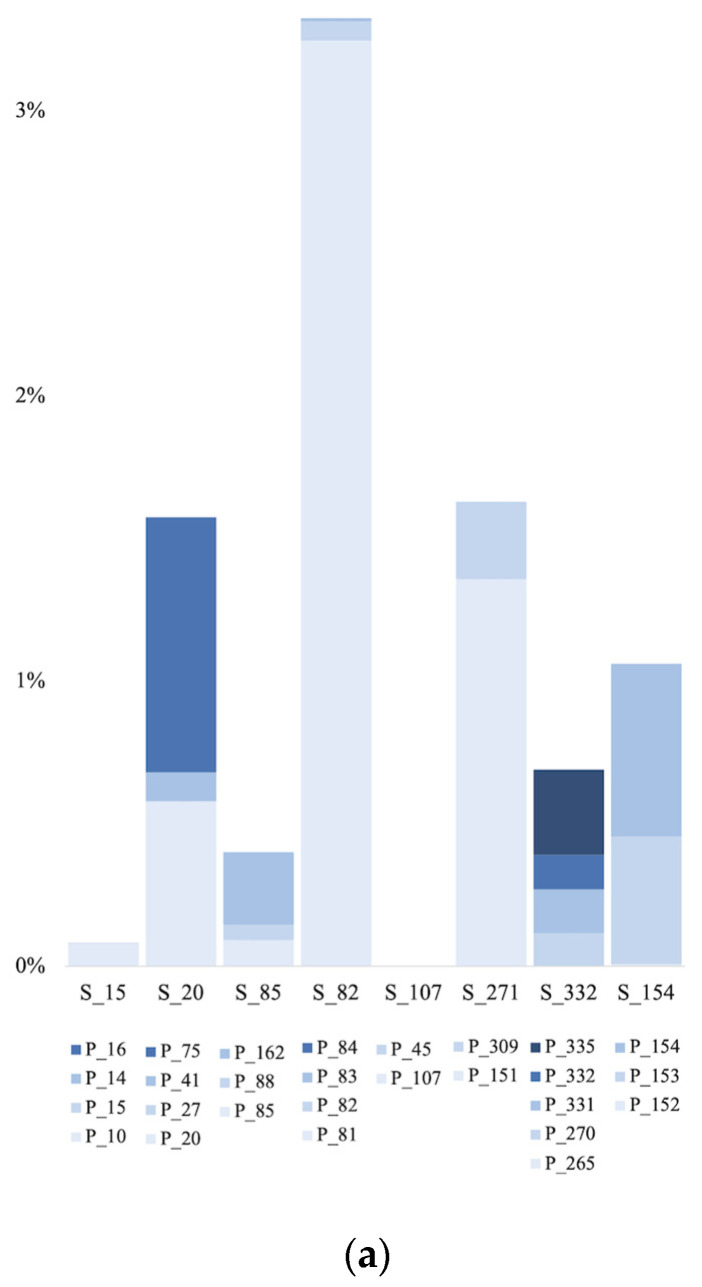
Archaeal (**a**), bacterial (**b**), and eukaryotic (**c**) community source tracking for surface habitats, using fast expectation-maximization microbial source tracking (FEAST). P, groundwater samples; S, surface water samples.

**Table 1 microorganisms-11-01674-t001:** Variation in archaeal, bacterial, and eukaryotic community composition explained by habitat (surface or subsurface), tested using PERMANOVA.

	Df	SumOfSqs	R2	F	Pr (>F)
ARCHAEA					
Habitat	1	0.9035	0.04216	1.8927	0.001
Residual	43	20.5271	0.95784		
Total	44	21.4306	1.00000		
BACTERIA					
Habitat	1	1.7682	0.06323	3.8472	0.001
Residual	57	26.1979	0.93677		
Total	58	27.9662	1.00000		
EUKARYOTE					
Habitat	1	1.5954	0.06515	3.5542	0.001
Residual	51	22.8929	0.93485		
Total	52	24.4883	1.0000		

**Table 2 microorganisms-11-01674-t002:** Correlation between community composition and explanatory factors using db-RDA. DOC, dissolved organic carbon; DIC, dissolved inorganic carbon.

	Df	SumOfSqs	F	Pr (>F)
ARCHAEA				
DOC	1	0.5443	1.1176	0.004
DIC	1	0.6213	1.2758	0.001
Residuals	42	20.4538		
BACTERIA				
DOC	1	0.6047	1.2629	0.006
DIC	1	0.8119	1.6956	0.001
Residuals	56	26.8145		
EUKARYOTE				
DOC	1	0.6600	1.4583	0.007
DIC	1	0.9515	2.1026	0.001
Residuals	50	22.6271		

## Data Availability

The obtained sequences were deposited in the National Center for Biotechnology Information (NCBI) under the BioProject ID PRJNA867732.

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
