# Peer review of "Linking Groundwater to Surface Discharge Ecosystems: Archaeal, Bacterial, and Eukaryotic Community Diversity and Structure in Quebec (Canada)"

_microorganisms, 2023, doi:10.3390/microorganisms11071674_

Round 1

Reviewer 1 Report

Linking groundwater to surface discharge ecosystems: Archaeal, Bacterial, and Eukaryotic community diversity and structure in Quebec (Canada)

The authors used amplicon sequencing to look at the diversity and organization of the archaeal, bacterial, and eukaryotic communities in groundwater and surface water samples from the vast Laurentides and Lanaudières regions of Quebec province, Canada. It may be helpful to microbial source tracking because in the current study only a small portion of the microbial communities in the groundwater contributed to the surface discharge communities. The paper is generally well-written but the methodology framework is unclear and lacks potential references. 

The introductory section of this article lacks critical and in-depth analysis of the available literature and comprehensive relevant work.

Comment 1: As it is a global issue, therefore, authors should incorporate quantitative data about the aquifers systems of different countries, and it would be good to cite and relate some other studies that investigated similar aspects.

Comment 2: I suggest you add 2-3 introductory sentences about Laurentides and Lanaudière regions. The reason to select these aquifers should be explained.

Comment 3: Explain the novelty of the work in a paragraph and make a comparison with the literature. There is a major gap to link previous work and the planned objectives of this work.

Comment 4: Besides the wide use of aquifers systems, how about its contribution to the natural environment and current water resources?

Comment 5: Detailed information should be given about soil sample collection, preservation, and analytical techniques.  There is no reference given in the materials and methods section

https://doi.org/10.1016/j.cej.2019.123674 https://doi.org/10.1016/j.watres.2019.03.079

Comment 7: I suggest you incorporate longitude and latitude data of sampling points and describe them in brief.

Comment 8: There are several citations in methods that should be replaced with primary references.

Comment 9: Authors should precisely focus on and strengthen this section with published research work.

Comment 10: I suggest you conduct a multivariate statistical analysis and determine the interaction effect among key parameters of this study.

Comment 11: In the discussion section, the ecological significance is not pointed out clearly by the authors.

Comment 12: In order to meet the requirements for publication, the quality of figures and tables should be improved.

Comment 13: Please rewrite this section with a prime focus on your key findings. 

There are numerous low-level errors in the text (including but not limited to unit symbols, punctuation marks, font sizes, and paragraph structure…). Please check the format of the text throughout this manuscript and unified its pattern. Correcting these errors should not be the responsibility of the reviewers. Please check it carefully throughout this manuscript. 

Author Response

We thank the reviewer for taking the time to read our paper, and for helping us improve it.

It may be helpful to microbial source tracking because in the current study only a small portion of the microbial communities in the groundwater contributed to the surface discharge communities.

Answer: We do not understand this comment? Microbial source tracking was carried out in this study (Figure 8).

The paper is generally well-written but the methodology framework is unclear and lacks potential references.

Answer: We acknowledge this comment, however it would be very helpful if the reviewer would give us more details on what is lacking. We provided references for the geochemical analyses, DNA extraction, sequencing, bioinformatics and statistical analyses.

The introductory section of this article lacks critical and in-depth analysis of the available literature and comprehensive relevant work.

Answer: In the introduction L44-61 we survey studies carried out for submarine groundwater discharge and explain that there are very few studies focusing on freshwater habitats. We have cited the few studies that we did find. We would be grateful if the reviewer could give us some hints as to what could be added to improve this section.

Comment 1: As it is a global issue, therefore, authors should incorporate quantitative data about the aquifers systems of different countries, and it would be good to cite and relate some other studies that investigated similar aspects.

Answer: As mentioned in our response to the previous comment, as there is very little information regarding discharge of groundwater in freshwater habitats, we do not see what other data can be added in the introduction.

Comment 2: I suggest you add 2-3 introductory sentences about Laurentides and Lanaudière regions. The reason to select these aquifers should be explained.

Answer: We added information L70-75.

Comment 3: Explain the novelty of the work in a paragraph and make a comparison with the literature. There is a major gap to link previous work and the planned objectives of this work.

Answer: The novelty of the work was highlighted in the introduction L81-87.

Comment 4: Besides the wide use of aquifers systems, how about its contribution to the natural environment and current water resources?

Answer: We added a sentence in the introduction L85-87.

Comment 5: Detailed information should be given about soil sample collection, preservation, and analytical techniques.  There is no reference given in the materials and methods section 

https://doi.org/10.1016/j.cej.2019.123674 https://doi.org/10.1016/j.watres.2019.03.079

Answer: We do not understand this request, as soil samples were not collected in this study, only water.

Comment 7: I suggest you incorporate longitude and latitude data of sampling points and describe them in brief.

Answer: Since this biogeographical survey was part of a wider survey on groundwater resources, we cited the report where all this information can be found under reference #11.

Comment 8: There are several citations in methods that should be replaced with primary references.

Answer: We would be grateful if the reviewer could please let us know precisely which references would need to be replaced, and suggest which ones are more relevant. The references we used in this study are ones we commonly use.

Comment 9: Authors should precisely focus on and strengthen this section with published research work.

Answer: Could the reviewer please either indicate which section this comment is referring to, or give us line numbers?

Comment 10: I suggest you conduct a multivariate statistical analysis and determine the interaction effect among key parameters of this study.

Answer: PERMANOVA and db-RDA were conducted in this study. Could the reviewer please tell us which other multivariate analysis would be needed to add to our story?

Comment 11: In the discussion section, the ecological significance is not pointed out clearly by the authors.

Answer: This aspect is covered in the discussion L631-634, 638-643, 672-675, 681-682, 746-747, 774-777, and in the conclusion paragraph.

Comment 12: In order to meet the requirements for publication, the quality of figures and tables should be improved

Answer: We had to decrease the quality of the figures to be able to upload the manuscript upon submission. We did however also upload higher resolution tiff files for all figures. I suppose the editors could give the reviewer access to these files.

Comment 13: Please rewrite this section with a prime focus on your key findings. 

Answer: Could the reviewer please either indicate which section this comment is referring to, or give us line numbers?

There are numerous low-level errors in the text (including but not limited to unit symbols, punctuation marks, font sizes, and paragraph structure…). Please check the format of the text throughout this manuscript and unified its pattern. Correcting these errors should not be the responsibility of the reviewers. Please check it carefully throughout this manuscript.

Answer: We modified errors throughout the text. However, the format is imposed by a template in order to submit to this journal. As such, most of the formatting (text, figures, and tables) could not be modified.

Reviewer 2 Report

Thank you for giving me the opportunity to revise the MS entitled “Linking groundwater to surface discharge ecosystems: Archaeal, Bacterial, and Eukaryotic community diversity and structure in Quebec (Canada)” by Benjamin Groult and his/her colleagues that was submitted to “Microorganisms”. The MS submitted is suitable for Microorganisms, and some interesting results were showed. However, there are several suggestions that have to consider by the authors. The Abstract section needs to be carefully revised. The innovation of the manuscript must be clearly stated in the text. Please pay attention to the format of the entire text.

Author Response

We thank the reviewer for taking the time to read our paper, and for helping us improve it.

The Abstract section needs to be carefully revised.

Answer: We acknowledge this comment, however it would be very helpful if the reviewer would give us more details on what (which parts?) need to be revised.

The innovation of the manuscript must be clearly stated in the text.

Answer: The novelty of the work was highlighted in the introduction L81-87.

Please pay attention to the format of the entire text.

Answer: We modified errors throughout the text. However, the format is imposed by a template in order to submit to this journal. As such, most of the formatting (text, figures, and tables) could not be modified.

Reviewer 3 Report

Your manuscript focuses on the microbial communities between the surface and groundwater ecosystems. Your work could contribute for a better knowledge of the link between these two communities. In general, the manuscript is organized and easy to read, albeit having many information as supplementary materials.

Regarding the eukaryote community, only the microeucaryote taxa should be presented, but you have many that do not belong to this community, please remove them, and re-analyze the data.

General remarks

- It is a pity that only two environmental parameters were assessed in the samples (DOC and DIC).

- Since the manuscript was submitted to a publication in the microbiology area, I think the eukaryotic community should referred only to the eukaryote microorganisms (microeucaryote), which is not the case. The α-, β-diversities indices and all the analyses involving of the microeucaryote community should be recalculated after remove the non-microbial taxa. Correct also Figs S3A and S3B, removing all the non-microbial taxa (even DNA Vertebrata is presented!).

I am convinced that your major findings will be the same, but for the accuracy of the manuscript, these corrections must be done.

- Some of the many supplement figures should be in the manuscript, as merged figures (the ones with the raw data – the bar graphics with the three communities in the samples. Also, some figures (a, b, c, in the manuscript) can be merged in one, with only one caption.

- Regarding the groundwater, do you have the sampling depth? Some

- In fig. S3A and S3B, several eukaryotic taxa are not microorganisms. Why not remove them from the figure?

- correct some taxa names; What do you mean by Phragmaplastophyta? with Phragmaplast?

- From the eucaryota list of the microbial community, please remove “Pinus”, and “unc. Embryophyta”, and many others.

- In fig 4 a,b,c, - try to enhance the area of the graphs with the communities dispersion.

- Suggestions: cluster analysis of the samples – for each community, and with all. The cluster analysis could be combined with the Principal coordinate analysis (PCoA).

Author Response

We thank the reviewer for taking the time to read our paper, and for helping us improve it.

Regarding the eukaryote community, only the microeucaryote taxa should be presented, but you have many that do not belong to this community, please remove them, and re-analyze the data. 

- Since the manuscript was submitted to a publication in the microbiology area, I think the eukaryotic community should referred only to the eukaryote microorganisms (microeucaryote), which is not the case. The α-, β-diversities indices and all the analyses involving of the microeucaryote community should be recalculated after remove the non-microbial taxa. Correct also Figs S3A and S3B, removing all the non-microbial taxa (even DNA Vertebrata is presented!). 

I am convinced that your major findings will be the same, but for the accuracy of the manuscript, these corrections must be done.

- In fig. S3A and S3B, several eukaryotic taxa are not microorganisms. Why not remove them from the figure?

- correct some taxa names; What do you mean by Phragmaplastophyta? with Phragmaplast?

- From the eucaryota list of the microbial community, please remove “Pinus”, and “unc. Embryophyta”, and many others.

Answer: As all these comments are related, we will answer them all at once. Indeed, this was a nice suggestion, and we reran all analyses with an updated eukaryotic dataset. The deleted taxa are mentioned in the methods L164-167. The updated analyses resulted in the same conclusions, except for some unique taxa that were not all the same as in the initial submitted version. However, these new taxa do not change the core results nor discussion and all modifications can be found in the results and discussion sections.

General remarks

- It is a pity that only two environmental parameters were assessed in the samples (DOC and DIC).

Answer: We completely agree with the reviewer, however this was out of our hands as we collaborated with a hydrogeology team who was leading this province-wide project on groundwater assessment. We only had a limited amount of time to sample, and limited resources at our disposal for sample collection.

- Some of the many supplement figures should be in the manuscript, as merged figures (the ones with the raw data – the bar graphics with the three communities in the samples. Also, some figures (a, b, c, in the manuscript) can be merged in one, with only one caption.

Answer: We thank the reviewer for this comment and have integrated the taxonomy figures in the text (now Figures 2, 3 and 4). We deleted the redundant figure legends but did not decrease the size of the figures to keep some quality to them (Figures 2, 3, 4 and 5). We are certain this will be addressed at the production stage, should our paper be accepted for publication.

- Regarding the groundwater, do you have the sampling depth? Some

Answer: Since this biogeographical survey was part of a wider survey on groundwater resources, we cited the report where all this information can be under reference #11.

- In fig 4 a,b,c, - try to enhance the area of the graphs with the communities dispersion.

Answer: We apologize but we do not understand this comment?

- Suggestions: cluster analysis of the samples – for each community, and with all. The cluster analysis could be combined with the Principal coordinate analysis (PCoA).

Answer: We thank the reviewer for this suggestion. However, we do not believe adding another figure to an already loaded figure would add anything relevant to the text nor the discussion. Community distances have been analyzed and discussed using the PCoA graphs.

Round 2

Reviewer 3 Report

I appreciate your efforts to improve the manuscript.